# Compression Performance and Deformation Behavior of 3D-Printed PLA-Based Lattice Structures

**DOI:** 10.3390/polym14051062

**Published:** 2022-03-07

**Authors:** Dongxue Qin, Lin Sang, Zihui Zhang, Shengyuan Lai, Yiping Zhao

**Affiliations:** 1Department of Radiology, The Second Affiliated Hospital of Dalian Medical University, Dalian 116027, China; 17709875038@163.com (D.Q.); drlsy@126.com (S.L.); 2School of Automotive Engineering, Dalian University of Technology, Dalian 116024, China; sanglin@dlut.edu.cn (L.S.); zhangzihui@mail.dlut.edu.cn (Z.Z.)

**Keywords:** polylactic acid (PLA), compression properties, triply periodic minimal surfaces (TPMS), computed tomography (CT) scanning

## Abstract

The aim of this study is to fabricate biodegradable PLA-based composite filaments for 3D printing to manufacture bear-loading lattice structures. First, CaCO_3_ and TCP as inorganic fillers were incorporated into a PLA matrix to fabricate a series of composite filaments. The material compositions, mechanical properties, and rheology behavior of the PLA/CaCO_3_ and PLA/TCP filaments were evaluated. Then, two lattice structures, cubic and Triply Periodic Minimal Surfaces-Diamond (TPMS-D), were geometrically designed and 3D-printed into fine samples. The axial compression results indicated that the addition of CaCO_3_ and TCP effectively enhances the compressive modulus and strength of lattice structures. In particular, the TPMS-D structure showed superior load-carrying capacity and specific energy absorption compared to those of its cubic counterparts. Furthermore, the deformation behavior of these two lattice structures was examined by image recording during compression and computed tomography (CT) scanning of samples after compression. It was observed that pore structure could be well held in TPMS-D, while that in cubic structure was destroyed due to the fracture of vertical struts. Therefore, this paper highlights promising 3D-printed biodegradable lattice structures with excellent energy-absorption capacity and high structural stability.

## 1. Introduction

Lattice structures have attracted increased attention for their lightweight effect, high specific stiffness–strength, geometry designability, and excellent energy-absorption capacity [1,2]. Cellular unit topography plays a crucial role in the mechanical performance of lattice structures [3,4]. Initially, some complex lattice structures failed to be manufactured due to the limit of processing and manufacturing, and the mechanical properties and failure modes were studied via finite-element analysis [5]. Additive manufacturing (AM) has opened the door for the realization of designed lattice structures [6,7]. Great efforts have been made in the preparation of lattice structures via different 3D printing techniques. Choy et al. [8] fabricated of two kinds of lattice structures using the selective laser melting (SLM) technique with Ti-6Al-4V as the building material, and explored the compression properties. Wang et al. [9] designed and constructed an octet truss topological scaffold on matched natural cancellous bones using the stereo lithography appearance (SLA) technique. Additionally, Antony et al. [10] fabricated hemp fiber reinforced polylactic acid feedstock filaments and then printed honeycomb sandwich structures with superior mechanical performance using the fused deposition modeling (FDM) technique, which showed a lot potential in industry, especially in the manufacture and design of automotive and aerospace prototypes.

Among the above-mentioned 3D printing techniques, FDM has the advantage of cost-effectiveness combined with a high degree of customization [11,12]. A typical FDM system involves the extrusion of molten thermoplastics through a heating nozzle and then movement of the nozzle on a build-plate bed along a predetermined path. It is worth mentioning that FDM is a term trademarked by Stratasys, and this process is synonymous with Fused Filament Fabrication (FFF), which is defined by ISO/ASTM (2015) as material extrusion [13]. Therefore, various thermoplastic polymeric composite feedstocks can be realized using FFF for additive manufacturing, which has led to an increased availability and use within industry. Polylactide acid (PLA) has been widely used as fused deposition modeling (FDM) material due to its excellent printability, mechanical properties, and eco-friendliness [14]. Additionally, PLA has been extensively applied as biomedical material due to its biodegradability and biocompatibility [15,16]. The incorporation of apatite fillers (i.e., hydroxyapatite (HAp)) could not only increase bioactivity but also enhance the load-carrying capacity of porous scaffolding structures. Therefore, PLA composites have great potential as FFF building material and the manufactured lattice structures as synthetic trabecular bone scaffolds.

Recent research has been carried out to fabricate PLA composite structures as bone scaffolds using FFF 3D printing [17,18,19]. Zhang et al. [19] 3D-printed four porous structures (square, triangle, honeycomb, and rhombus) of PLA/HA scaffolds with comprehensive performance optimizations to meet the requirements of personalized bone repair applications. Gendiviliene et al. [18] fabricated woodpile architectures of PLA/Hap and investigated the dimensional accuracy using FDM. The deformation behavior was mainly revealed through digital recording or FEA simulation [20,21]. However, few studies have focused on the fabrication of triply periodic minimal surface (TPMS) structures via FFF printing, whose structure possessed excellent energy absorption and high load-bearing capacity. The failure modes of 3D-printed PLA composite TPMS structures are still not known, and the evolution of internal pore architecture compression load is also seldom investigated, which is rather important for the biological response path in cells and new bone trabecular in-growth.

Therefore, the aim of this study is to assess the feasibility of the fabrication of TPMS lattice structures using lab-made PLA composite filaments. Bioactive inorganic fillers, including calcium carbonate (CaCO_3_) and calcium phosphate (TCP), were added into a PLA matrix. The influence of material compositions and cellular geometry on compression performance and energy absorption was investigated. More importantly, the deformation behavior of pore architecture was examined using computed tomography (CT) scanning and reconstruction.

## 2. Experimental

### 2.1. Materials

Granular poly(lactic acid) (PLA, REVODE 195, Zhejiang Hisun Biomaterials Co. Ltd. Taizhou, China), was used as the matrix as received. Calcium carbonate (CaCO_3_) and tricalcium phosphate (TCP) powder with average size of 1 μm was purchased from Zhongshan Techwill Trading Co., Ltd., Zhongshan, China.

### 2.2. Preparation of PLA/CaCO_3_ and PLA/TCP Composite Filaments for 3D Printing

PLA pellets and mineral salts (including CaCO_3_ and TCP) were fully dried and then blended by a twin-screw extruder (Nanjing Giant Machinery Co., Ltd., Nanjing, China) with a ratio of L/D = 40. PLA and a varying content of inorganic fillers (5, 10 and 20 wt%) were premixed by high-speed mixer, and then fed into the extruder under the following temperature profile as 170, 175, 180, and 180 °C. The extruded strand was graduated into pellets and dried at 80 °C for 24 h before further processing. The material compositions and code method are listed in Table 1.

PLA-based composite filaments (1.75 ± 0.1 mm diameter) were extruded using prepared pellets through a single-screw filament extruder (Wellzoom C, Shenzhen Mistar Technology Co., Ltd. Shenzhen, China). The extruder barrel heating zone to die temperatures were set at 205 and 210 °C, respectively. The schematic preparation route of PLA-based composte filaments were illustrated in Figure 1, and the fabricated composite filaments were dried and stored in vacuum-sealed bags prior to 3D printing.

### 2.3. Lattice Structure Model

In the current work, two lattice structures were chosen to construct the printed models (Figure 2). First, the cubic structure of a 3D-printed scaffold was modeled using CAD software (CATIA V5) and then exported to STL files. The scaffold struts were fixed at 0.5 mm, and different pore sizes of 500, 650, and 800 μm were designed. Then, the STL files were imported to the CURA program for slicing and the setting of printed parameters.

Second, triply periodic minimal surfaces (TPMS) structures were mathematically approximated using implicit methods [22,23]. Among different types of TPMS structures, diamond (D) surface was selected in the current work. The D-surface was described as follows:(1)$$\begin{array}{c} \phi D(x,y,z)=\sin(\omega )\sin(\omega y)\sin(\omega z)+\cos(\omega x)\sin(\omega y)\sin(\omega z)+\\ \sin(\omega x)\cos(\omega y)\sin(\omega z)+\sin(\omega x)\sin(\omega y)\cos(\omega z)=C \end{array}$$
where *x*, *y*, and *z* represent spatial coordinates, *ω* = 2π/l and l is the length of a unit cell. The D-surface was generated as the solution of a level-set function *ϕ* = *C*. The solid model of diamond surfaces was created by extracting the zero-level-set surface (when *C* = 0) from Equation (1). Matlab scripting was used to generate the sheet surfaces. The resultant 3D STL models were then transferred to CURA software for slicing in preparation for 3D printing (Figure 2).

### 2.4. D Printing of Test Specimen and Lattice Structures

Three-dimensionally printed tensile samples and lattice structures were produced using an FDM printer (Ultimaker 2+, Ultimaker Manufacture, Utrecht, The Netherlands). The printing parameters was set as follows: the nozzle diameter of 0.40 mm, layer height of 0.15 mm, the infill density of 100%, and the printing velocity of 20 mm/s. For PLA/CaCO_3_ and PLA/TCP composite filaments, the nozzle temperature was 210 °C, and the building platform temperature was 60 °C.

### 2.5. Compositional, Rheology, and Micro-Structure Characterization

The chemical composition of the filaments was identified by X-ray diffraction diffractometer (XRD, D8, Bruker) with a copper radiation source (Cu Kα, λ = 1.5406 Å) and a secondary monochromator operated at 40 kV and 40 mA over a range from 4° to 60°.

Rheological testing was carried out in the strain-controlled rheometer (DHR-2, TA-Instruments). The samples were 3D-printed into parallel plates 25 mm in diameter and 2 mm in thickness. Storage modulus and complex viscosity was collected during testing at the testing temperature of 210 °C.

The morphological features of the FDM-printed scaffolds were observed by scanning electron microscope (SEM, QUANTA 450, FEI). All the printed cellular structures were gold-sputtered before observation.

### 2.6. Mechanical Test

The tensile properties of FDM-printed dog-bone samples were assessed using a universal mechanical testing machine (GT-7001-HC6, 20KN, GOTECH TESTING MACHINE INC, Taiwan, China) at a constant crosshead speed of 2 mm/min at ambient temperature according to the ISO 527 standard. Each specimen was tested for five dumbbell replicates.

The stiffness of the FDM-printed scaffolds was evaluated from compression tests, according to the ASTM D-695 standard. Compression properties were performed using a universal machine testing machine (TSE 105D, 100KN, Shenzhen Wance Test Equipment Co., Ltd., Shenzhen, China). Unconstrained cellular structure samples with 12.7 mm × 12.7 mm × 25.4 mm (length × width × height) were compressed between flat steel plates at a constant strain rate of 2 mm/min. Deformation of samples during compression testing was captured in photograph images by an advanced contactless charge-coupled device (CCD, Huagu Power Technology, Co., Ltd., Shenzhen, China). The image was recorded at one frame every two seconds. Tests were terminated at a strain of 20% of lattice structures. Each specimen was tested for at least three replicates for compression tests.

The energy absorption (*EA*) of the lattice structures during the compression process was calculated from the area under the stress–strain curve as follows (Equation (2)) [24]:(2)$$EA=\overset{b}{\underset{a}{\int }}\sigma \cdot d\varepsilon \;$$
where *σ* assigns to the compressive stress and the *ε* is the nominal strain. The calculation of *σ* = F/A and *ε* = δ/H, respectively. F and δ correspond to the compressive force and displacement, which are recorded during the compression test. A is the original cross-section area and H is the height of the D-surface TPMS structure along the compressive direction. The specific energy absorption (SEA) is obtained by *EA* regarding the sample weight.

### 2.7. Computed Tomography

The internal structure of lattice samples after the compression test was scanned and reconstructed by computed tomography (CT equipment, SIEMENS SOMATOM DRIVE, Germany). The CT scan voltage was 70 kV with a current of 61 mA. Each slice thickness was set at 0.5 mm with a slice spacing of 0.3 mm. The field of view (FOV) was 50 mm, and the matrix pixel was 512 × 512. The dose length product (DLP) was approximately 45.58 mGy. The reconstruction based on the obtained data used the SIMENS software analysis system (Syngo CT VA62A).

## 3. Results and Discussion

### 3.1. Material Characterization

The preparation route of PLA-based composite filaments for 3D printing is shown in Figure 1. Varying contents of inorganic fillers, including CaCO_3_ and TCP, were incorporated into the PLA matrix. Uniform and standard filaments were achieved, and tensile specimens were successfully FDM-printed without defects. XRD patterns were used to confirm the addition of inorganic fillers. In Figure 3, the diffraction patterns for pure PLA, TCP, CaCO_3_ and a series of composite materials are listed. It is possible to observe that neat PLA exhibited a broad diffraction peak centered at 2θ ~ 18°, indicating the pure crystalline characteristics of PLA. After being filled with CaCO_3_ or TCP particles, the broad hump from the XRD pattern of both PLA/CaCO_3_ and PLA/TCP became sharp and distinct. Specifically, peaks at 16.8° and 19.2°, corresponding to crystal planes of (110/200) and (203) assigned to the PLA polymer, were detected in the XRD patterns of composite filaments, which indicated a nucleation effect after the incorporation of inorganic fillers. Moreover, characteristic peaks of inorganic fillers were also observed. For example, strong peaks of CaCO_3_ at 29.5°, 35.9°, and 39.4°were detected in composite filaments [25], confirming the successful incorporation of CaCO_3_.

The effect of the addition of CaCO_3_ and TCP on the tensile behavior of the composite filament is presented in Figure 4. For the series of PLA/CaCO_3_ filaments, 5% and 10% addition increased the tensile strength from 54.7 MPa (pure PLA) to 62.5 MPa (PLA/CaCO_3_5%) and 61.0 MPa (PLA/CaCO_3_10%), respectively. However, when the CaCO_3_ content reached 20%, tensile strength declined to 49 MPa, which was 19% lower than pure PLA. Meanwhile, the ductility decreased, and the stiffness (modulus) was enhanced after the addition of CaCO_3_. On the other hand, the incorporation of 5% TCP maintained the tensile property as pure PLA, while content of 10% TCP slightly weakened the tensile strength of the composites. When the inorganic fillers reached 20%, it was observed that tensile strength became deteriorated. This might be due to the aggregation of filler particles and induced stress concentration, which is further discussed regarding the surface morphology of lattice structures.

The flowability of fabricated composite filaments was closely related to nozzle extrusion during 3D printing. Therefore, rheological tests were conducted in PLA/CaCO_3_ and PLA/TCP with varying filler contents, and the storage modulus (G′) and complex viscosity on frequency are shown in Figure 5. Similar rheological behavior in storage and loss modulus of PLA/CaCO_3_ and PLA/TCP materials was observed. With inorganic filler increased, the storage moduli were all increased, suggesting enhancement of the modulus. The complex viscosity of PLA/CaCO_3_5%, PLA/CaCO_3_10%, PLA/TCP5% and PLA/TCP10% was kept relatively constant from 1 to 100 rad/s, suggesting stable rheological behavior during the extrusion process. However, complex viscosity was relatively high and largely fluctuated in PLA/CaCO_3_20% and PLA/TCP20% material, which required more energy to extrude melt flow and deteriorated the mechanical property in the FDM-printed specimens.

### 3.2. D-Printed PLA-Based Lattice Structures

Previous research has proved that TPMS structures (i.e., Diamond, Gyroid, etc.) exhibited excellent compression performance and energy-absorption capacity [26]. Therefore, the cubic and TPMS-D lattice structure was adopted and then 3D-printed using fabricated PLA/CaCO_3_ and PLA/TCP composite filaments. The geometry designs of the two kinds of lattice structures are illustrated in Figure 6. When applied as bone implants, mechanical performance and the migration/proliferation of cells were dependent on the pore size of the scaffold [27]. Thus, the pore size of the cubic structure was set at 500, 650, and 800 μm, respectively. According to the designed models, PLA-based cubic and TPMS-D structures were successfully printed with a fine appearance.

Furthermore, Figure 7 presents the representative SEM images of two kinds of printed models in the top-side view. It is clear that the both the PLA/CaCO_3_ and PLA/TCP TPMS-Diamond models show a porous structure with structural agreement in the presented sections, suggesting a feasibility of fabricating lattice structures via FDM and the excellent FDM printability of our lab-made filaments. In magnified micrographs, the distribution of inorganic fillers (CaCO_3_ and TCP) becomes evident with the increase of mass fraction. When the addition of CaCO_3_ and TCP was 5%, a relatively smooth surface with few particles exposed was obtained in the printing layers. However, with increased filler content, some particles were detected on the surface printing layer. When the filler content reached 20%, local defects and clusters of filler particles were observed, which was associated with increased viscosity and decreased melt strength for PLA/CaCO_3_ 20% and PLA/TCP 20% composites. The local aggregation of ceramic particles is challenging for 3D printing, which might deteriorate mechanical and processing properties. Nevertheless, the increase of surface roughness would be beneficial for cell adhesion and migration during the tissue regeneration process.

The mechanical response to load-bearing is considered to be an important characteristic for scaffold structures [28]. The compression properties of PLA/CaCO_3_5% cubic lattice structures with varying pore sizes and filler content were investigated, and pure PLA was set as controls. It can be seen that all the compression stress–strain curves consisted of wave-period peaks until compression strain reached 20% (Figure 8). Each peak showed an initial increase in stress, a peak hit, and then a dramatic drop in stress. Although the material composition or pore size varied, the compressive response of cubic lattice structure was not greatly changed.

The compression performance, including modulus and peak strength, was extracted and listed in Figure 8. The compression modulus was achieved by the slope of the initial linear region of the stress–strain curve [29], whereas compression strength was set as the maximum value of the first peak in the stress–strain curves. All the 3D-printed PLA and PLA/CaCO_3_ scaffolds displayed a compression strength that belonged in the range of human trabecular bone (4–25 MPa) [27], confirming the great potential of cubic PLA-based lattice structures for use as bone implants. Meanwhile, the compression modulus was greatly increased after adding a small amount of CaCO_3_ filler, endowing the cubic lattice structure with an effective load-carrying capacity. However, no statistical difference in compression strength and modulus was found among the PLA and PLA/CaCO_3_ samples with different pore sizes. The compressive modulus ranged from 0.3–0.9 GPa, which was also suitable for the modulus of trabecular structure (0.01–1 GPa) [30]. Cubic scaffolds with 500 μm pore size possessed favorable compression performance, and thus a pore size of 500 μm was adopted in the following test.

The influence of filler content on the compression response of PLA/CaCO_3_ and PLA/TCP lattice structures was further investigated. In Figure 9a–d, the compression stress–strain curves of each lattice structure displayed similar variation trends, suggesting that the compression response was more related to cellular topography than material composition. However, the stress–strain curves of TPMS-D behaved stably and can be divided into two regions, which is quite different from the wave-shaped curves of the cubic structure. First, the stress increased linearly and exhibited an approximately linear stress–strain relationship, suggesting elastic deformation. Second, the stress increased slowly with plastic deformation. All the curves of the TPMS-D structure were regular and relatively stable, suggesting constant load-bearing during the compression test.

By comparing the cubic and TPMS-D samples, it was found that the TPMS-D structures resulted in a continuous improvement of compressive strength, and all the maximum values were above 30 MPa for PLA/CaCO_3_ and 20 MPa for PLA/TCP samples. In particular, the compressive strength reached 50 MPa for the PLA/CaCO_3_10% TPMS-D sample. By contrast, the compression strength only ranged from 13–20 MPa. Consequently, the TPMS-D structure exhibited excellent load-bearing capacity in comparison to the cubic lattice structure. Moreover, one should note the particular case of the PLA/TCP20% TPMS-D structure, where a sharp drop was observed because of the deteriorated mechanical properties of the prepared filament. For both kinds of structure, PLA/CaCO_3_10% and PLA/TCP10% filaments exhibited better compressive performance.

Specific energy absorption (SEA) was another important factor for lattice structures. Therefore, the SEA of PLA/CaCO_3_ and PLA/TCP structural samples (cubic and TPMS-D) was further considered (as shown in Figure 10). It can be seen that both material composition and lattice structure had an impact on SEA values. Concerning the lattice structure, all TPMS-D structures exhibited significantly higher SEA values than those of their cubic counterparts. Specifically, the SEA values of the TPMS-D structures were >300% of cubic sample for PLA/CaCO_3_, and ~200% for PLA/TCP samples. On the other hand, PLA/CaCO_3_ filaments exhibited better compression performance than PLA/TCP filaments. Nevertheless, the structure design showed a more dominant contribution to the improvement of compression response.

The deformation behavior of the two kinds of lattice structures during the compression test were recorded at an applied nominal strain (0→0.2). It can be seen that abrupt shear failures were observed in the cubic structure, which occurred in a vertical strut in a random layer. This was because vertical struts bore the main compression load, and became unstable and were finally destroyed during compression. The phenomenon of abrupt shear failure in layers agreed with the occurred wave peaks in the stress–strain curves (Figure 11). Therefore, cubic structures might experience an unstable state when subjected to sustained compression stress. The internal porous structures were destroyed due to the vertical struts collapsing or fracturing, which might be a potential risk when used as implanted devices or tissue scaffolds. By contrast, no obvious damage was detected for TPMS-D structures. An outward expansion was seen in the middle part of the samples when subjected to compression load. Combined with the compression stress–stain curves, the deformation behavior of TPMS-D was preferred to a continuous accumulation process, suggesting a stable load-carrying capacity for TPMS-D structure.

Although photographs of the deformation process were continuously recorded by the camera, the interior damage and failure mode of the 3D scaffold structure was difficult to detect. Three-dimensional reconstruction images of lattice structures and the cross-sectional segments from bottom to top were further analyzed using computed tomography (CT) [31]. The grayscale histograms of four representatives of lattice structures after compression testing were CT-scanned and reconstructed, and their deformation behavior and pore structure in the internal structure were examined.

As shown in Figure 12a,b, damage was observed in PLA/CaCO_3_10% and PLA/TCP10% cubic strut structures. Some layers were separated (weaker visibility), while some layers were squeezed without pore space (sharper visibility) because of the fracture of vertical struts. The results of CT scans indicated that vertical struts played an important role in load-carrying and experienced a shear plane fracture under axial compression load. For separated regions, stress failed to transfer effectively. On the other hand, the squeezed region lost the porous structures, which presents potential risk when applied as implanted devices or tissue scaffolds. Additionally, the dislocation deformation of the whole sample was limited to a precise position. For the TPMS-D structure, a bulky plastic deformation without obvious fracture or crack damage was observed in both PLA/CaCO_3_10% and PLA/TCP10% samples (Figure 12c,d). In the cross-sectional CT scans, although the pore structure became distorted and deformed, the interconnection of the pore structure was well maintained. This phenomenon occurred mainly because the curved surface of the three-dimensional cellular unit possesses more plastic hinging under compression load [32]. The elastic–plastic deformation behavior can better withstand compression load and absorb more energy. Therefore, TPMS-D is a more promising lattice structure, with excellent compression performance, and the retention of interconnected pore structures is favorable in bone tissue engineering.

## 4. Conclusions

In this work, biodegradable PLA/CaCO_3_ and PLA/TCP composite filaments for fused a deposition modeling technique were developed. Two lattice structures, cubic and TPMS-D lattices, were designed and successfully 3D-printed. The compression performances of these two structures were further evaluated. The conclusions were as follows:

A series of PLA/TCP and PLA/CaCO_3_ composites filaments with 5–20 wt.% addition was prepared. Results showed that composite filaments with 5–10 wt.% addition displayed suitable tensile and rheology properties.

Cubic and TPMS-D structures exhibited a fine appearance, suggesting our lab-made filaments possessed excellent 3D printability. It was envisioned that TPMS-D exhibited superior load-carrying and energy-absorption capacities to those of cubic structures.

Computed tomography scanning showed different deformation behaviors for the cubic and TPMS-D structures. It was demonstrated that the failure mode of the cubic structure was mainly layer shear damage due to the fracture of vertical struts, while the TPMS-D sample exhibited a plastic deformation with an interconnected pore structure.

## Figures and Tables

**Figure 1 polymers-14-01062-f001:**
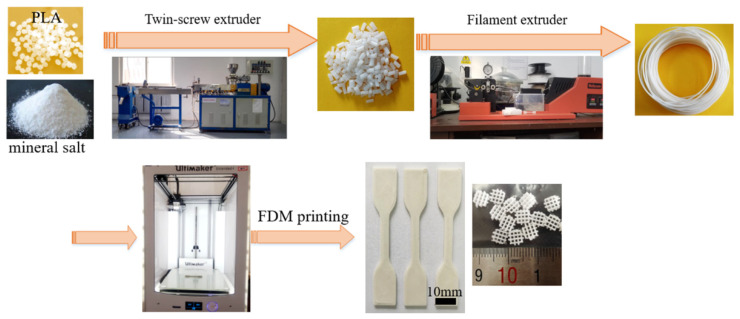
Schematic preparation route of PLA-based composite filaments and 3D-printed samples.

**Figure 2 polymers-14-01062-f002:**
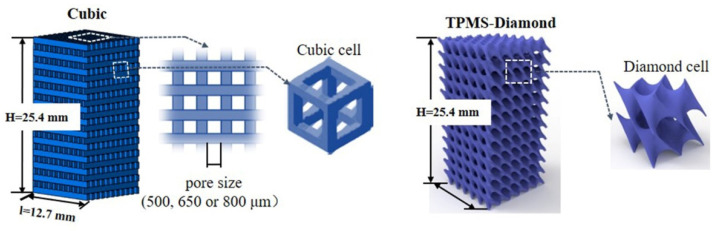
Geometry model of cellular structures of PLA composite specimens for compressive test (cubic cellular unit and TPMS-D cellular unit).

**Figure 3 polymers-14-01062-f003:**
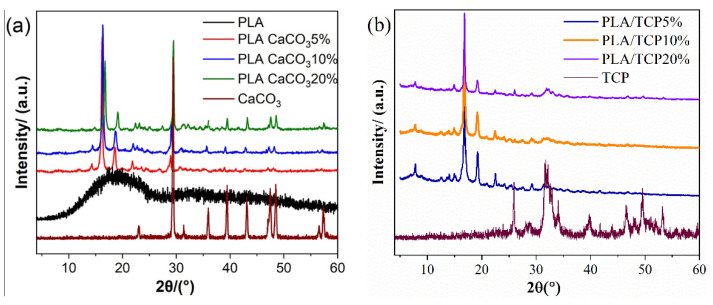
XRD patterns of (**a**) PLA/CaCO_3_ and (**b**) PLA/TCP composite filaments.

**Figure 4 polymers-14-01062-f004:**
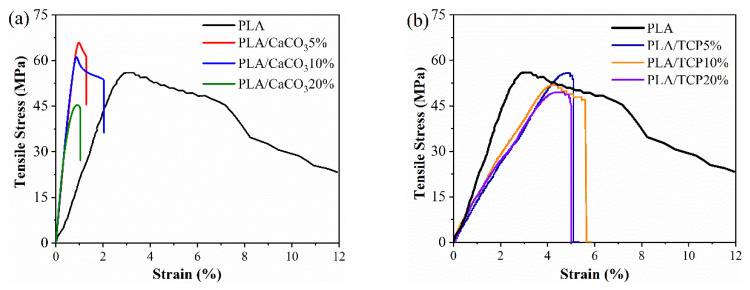
Tensile stress–strain curves of (**a**) PLA, PLA/CaCO_3_ and (**b**) PLA/TCP composite filaments.

**Figure 5 polymers-14-01062-f005:**
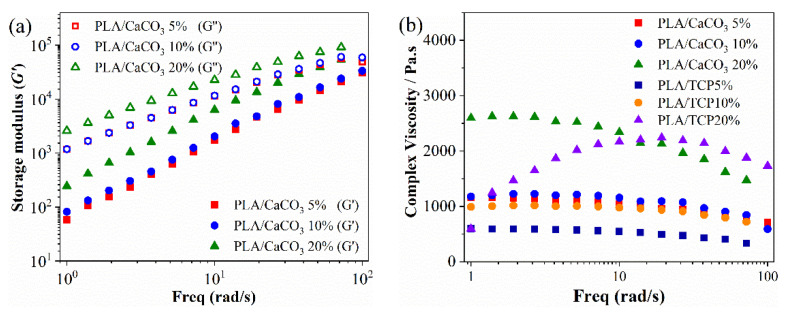
(**a**) Storage modulus and (**b**) the corresponded complex viscosity of PLA/CaCO_3_ and PLA/TCP composite materials.

**Figure 6 polymers-14-01062-f006:**
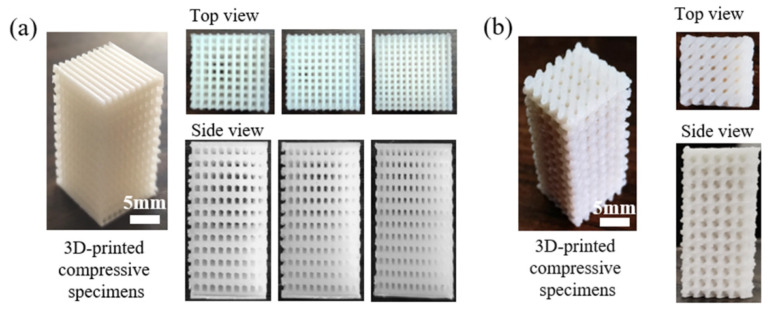
Digital photographs of 3D-printed (**a**) cubic and (**b**) TPMS-D specimens with top view and side view.

**Figure 7 polymers-14-01062-f007:**
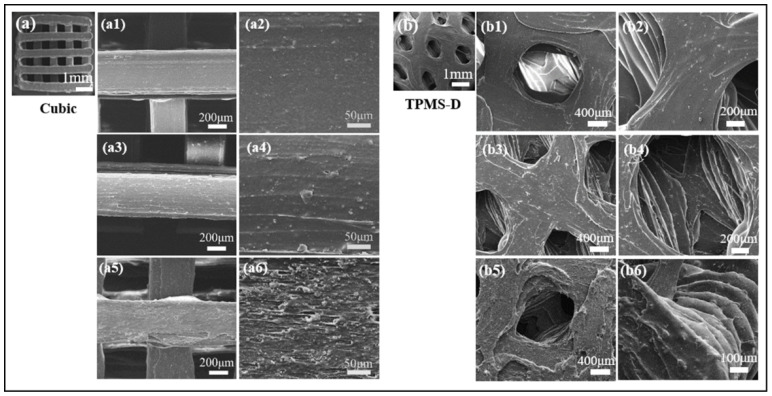
SEM images of 3D-printed lattice specimens of cubic- and TPMS-D lattice structures, (**a**) cubic structure: (**a1**) PLA/CaCO_3_5%, (**a2**) PLA/CaCO_3_10%, (**a3**) PLA/CaCO_3_20%, (**a4**–**a6**) were corresponded magnified images, (**b**) TPMS-D structure: (**b1**) PLA/TCP5%, (**b2**) PLA/TCP10%, (**b3**) PLA/TCP20% and (**b4**–**b6**) were the corresponded magnified images.

**Figure 8 polymers-14-01062-f008:**
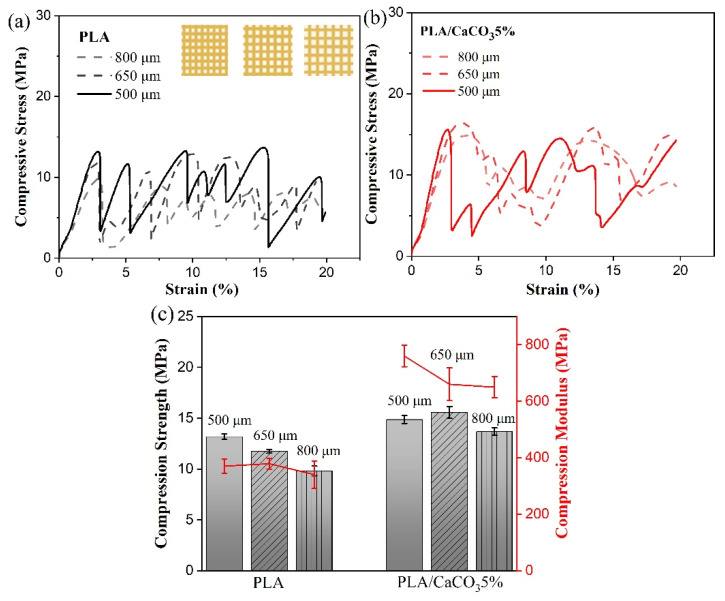
Compression stress–strain curves of (**a**) PLA, (**b**) PLA/CaCO_3_ cubic lattice structure with varying pore sizes, and histograms of (**c**) compression strength and modulus.

**Figure 9 polymers-14-01062-f009:**
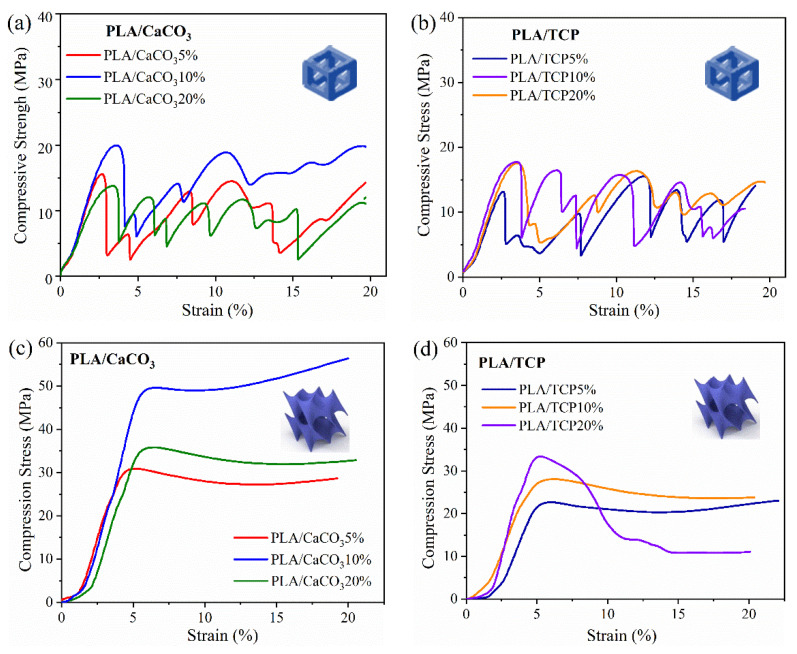
Typical stress–strain curves for FDM-printed structure subjected to compression test: (**a**) cubic PLA/CaCO_3_, (**b**) cubic PLA/TCP, (**c**) TPMS-D PLA/CaCO_3_ and (**d**) TPMS-D PLA/TCP.

**Figure 10 polymers-14-01062-f010:**
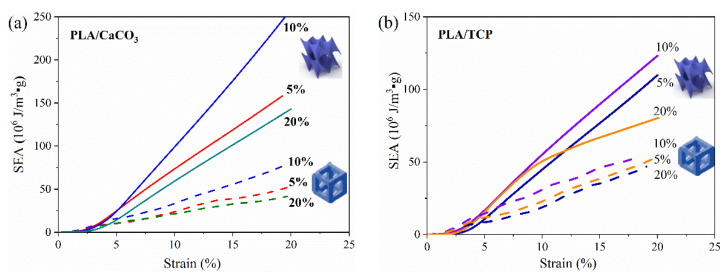
Specific energy absorption (SEA) of 3D-printed (**a**) PLA/CaCO_3_ and (**b**) PLA/TCP lattice structures.

**Figure 11 polymers-14-01062-f011:**
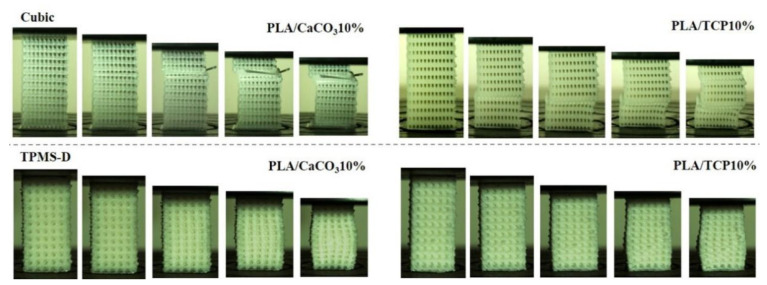
Macroscopic images of deformation of FDM-printed PLA/CaCO_3_10%, PLA/TCP10% cubic and TPMS-D lattice structures during the compression test.

**Figure 12 polymers-14-01062-f012:**
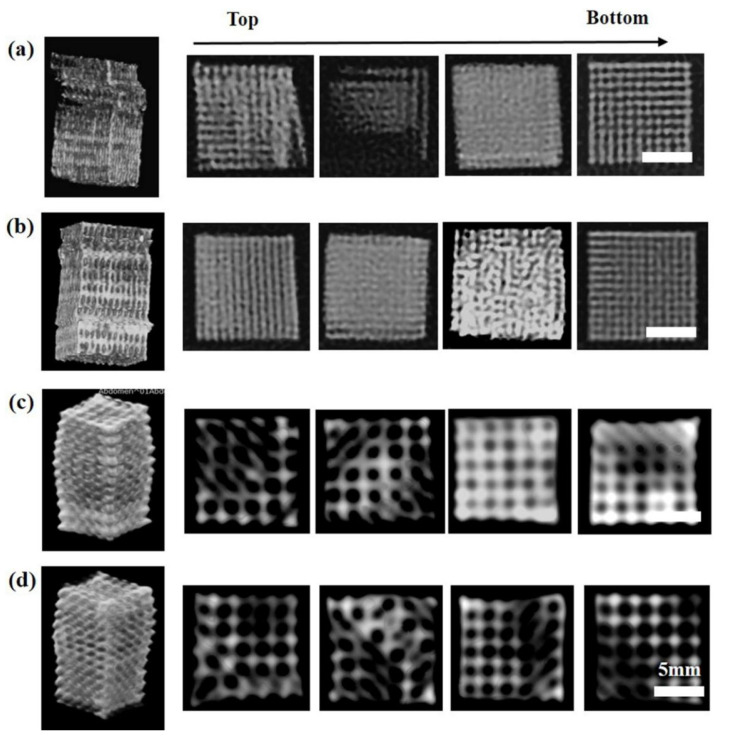
Three-dimensional reconstruction and local cross-section CT images of 3D-printed samples after compression test: (**a**) cubic PLA/CaCO_3_10%, (**b**) cubic PLA/TCP10%, (**c**) TPMS-D PLA/CaCO_3_10%, and (**d**) TPMS-D PLA/TCP10%.

**Table 1 polymers-14-01062-t001:** Composition of the prepared formulations for PLA and inorganic fillers.

Samples	PLA (wt%)	Inorganic Fillers (wt%)	Code
PLA/CaCO_3_	95	5	PLA/CaCO_3_5%
	90	10	PLA/CaCO_3_10%
	80	20	PLA/CaCO_3_20%
PLA/TCP	95	5	PLA/TCP5%
	90	10	PLA/TCP10%
	80	20	PLA/TCP20

## Data Availability

The data presented in this study are available on request from the corresponding author.

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
