# Peer review of "Compression Performance and Deformation Behavior of 3D-Printed PLA-Based Lattice Structures"

_polymers, 2022, doi:10.3390/polym14051062_

Round 1

Reviewer 1 Report

The authors studied the properties of lattice structures fabricated by fused filament fabrication using PLA reinforced with CaCO3 and TCP. While the manuscript is generally well executed, there are several issues that should be addressed before further consideration for publication.

  1. Suggest the authors to use ISO/ASTM terminology to describe the AM processes.
  2. Any characterisation on the composition of the filaments? For example, are the filler homogenously distributed after the extrusion?
  3. How many replicates are used for the experiments? For the graphs, are they the average values shown?
  4. Any discussion on the deviation between the samples fabricated and designed values? Do the fillers affect the results?

Author Response

Reviewer #1:

The authors studied the properties of lattice structures fabricated by fused filament fabrication using PLA reinforced with CaCO3 and TCP. While the manuscript is generally well executed, there are several issues that should be addressed before further consideration for publication.

Comment 1: Suggest the authors to use ISO/ASTM terminology to describe the AM processes.

Author’s reply 1: Thanks you. As you suggested, we have modified the AM process use the terminology words in 2.4 3D printing of test specimen and lattice structures as follows:

“3D-printed tensile samples and lattice structures were produced via a FDM printer (Ultimaker 2+, Ultimaker Manufacture, Netherlands). The printing parameters was set as follows: the nozzle diameter of 0.40 mm, layer height of 0.15 mm, the infill density of 100% and the printing velocity of 20 mm/s. For PLA/CaCO3 and PLA/TCP composite filaments, the nozzle temperature was 210 °C and the building platform temperature was 60 °C.”

AM process was introduced in a separate section to make the description more clearly.

Comment 2: Any characterization on the composition of the filaments? For example, are the filler homogenously distributed after the extrusion?

Author’s reply 2: Thank you. Before the fabrication of composite filaments, we carried out the melting flow index (MFI) to ensure the extrusion processibility. The MFI values of PLA/CaCO3 and PLA/TCP are ranged from 15-25 g/10(g â–ªmin), and the MFI increases with increasing content of inorganic fillers.

After achieving the filaments, we didn’t examine the state of filler distributed in the matrix. However, we printed some scaffolds (as shown in Fig. x1), and the surface microstructure and the compatibility between filler and matrix were observed by SEM (Fig. 7). We also modified SEM images to exhibit a better version of surface microstructure in the revision. Aggregated filler particles were observed when the filler content reached 20%, which indicated similar agglomeration occurred in the internal filament and inhomogeneous dispersion of particles

In addition, we also studied the thermal properties of the composite filaments via differential scanning calorimeter (DSC). We tested pure PLA, PLA/CaCO35%, PLA/CaCO310%, PLA/TCP5% and PLA/TCP10% composite filaments, and the DSC curves were shown in Fig. x2. After incorporation of inorganic fillers, the melting point increased form 168°C to 175°C, while the cold crystallization temperature decreased from 103°C to 95°C. This phenomenon suggested that the inorganic particles acted as nucleating agent and contributed to an increased crystallinity. The similar results were confirmed by the XRD patterns in Fig. 3, and we also added some discussion about this point in the revised manuscript.

Comment 3: How many replicates are used for the experiments? For the graphs, are they the average values shown?

Author’s reply 3: Thanks for your comment. For tensile specimens, each specimen was tested for five dumbbell replicates. After discarding curve with large deviation value, average values were shown in the graphs (at least from three curves). For example, tensile stress-strain curves of PLA/TCP5% and PLA/TCP20% were shown in Fig. x3, and the reproducibility between parallel samples seems good.

For lattice structures, at least three replicates were fabricated for compression tests. Due to the fracture of vertical struts of cubic structures, slight differences existed among the replicates (for example, Cubic-PLA/CaCO35%, Fig. x4). Nevertheless, the compressive strength and modulus were close. As you pointed out, we also added the replicates for the experiment in the revision.

Comment 4: Any discussion on the deviation between the samples fabricated and designed values? Do the fillers affect the results?

Author’s reply 4: Thanks for your comment. It is a very professional and good question. For FDM printing technique, the melting polymer extruded from a spherical heated nozzle and then cooled downed to form a solid part. Each layer was composed of a certain content of printing beads along the printing direction. The cross-section of printing beads were actually elliptic type (the following figure obtained from published paper (Compos Commun 22(2020) 100478) from co-author Dr. Sang). Accordingly, it is certain that there is a deviation between the samples and designed values).

    We tried to figure out the exact deviation between the printed sample and designed model, however, factors including printing parameters (infill density, infill line thickness, build-plate temperature, etc.), matrix type, filler incorporation and even the printing structure influence the deviation. Concerning on the fillers, the melt fluidity greatly influence the elliptic flatness of the cross-sectional printing beads. To reduce the impact of deviation, we compared the values of specific energy absorption (energy absorption per mass) in Fig. 10. Besides, we think porosity measurement and theoretical calculation might be a feasible way to achieve this deviation, and we will conduct this comparison in our future work.

    Once again, thank you very much for your thoughtful and thorough review. Hopefully, we have addressed all of your concerns. We look forward to receiving any information about our revision.

Reviewer 2 Report

Therefore, the aim of this study is to assess the feasibility of fabrication TPMS lattice 66 structures using lab-made PLA composite filaments. Bioactive inorganic fillers including calcium carbonate (CaCO3) and calcium phosphate (TCP) were added to PLA matrix. The influence of material compositions and cellular geometry on compression performance and energy absorption were investigated. The MS is interesting to readership. I suggest minor revisions.

  1. During the mechanical testing, at what newton force was applied?
  2. The XRD result of the composite materials needs to discuss more.
  3. The ratio of PLA or TCP or CaCO3 is not clear in manuscript. pleas mention clearly with weight mentioned as well.
  4. Authors emphasize fabrication of porous bioactive composite. Herein I suggest studying biocompatibility and cellular behavior on their product.

Author Response

Reviewer #2:

Therefore, the aim of this study is to assess the feasibility of fabrication TPMS lattice structures using lab-made PLA composite filaments. Bioactive inorganic fillers including calcium carbonate (CaCO3) and calcium phosphate (TCP) were added to PLA matrix. The influence of material compositions and cellular geometry on compression performance and energy absorption were investigated. The MS is interesting to readership. I suggest minor revisions.

Comment 1: During the mechanical testing, at what newton force was applied?

Author’s reply 1: Thank you. We used a universal mechanical testing machine (GT-7001-HC6, 20KN, GOTECH TESTING MACHINE INC, Taiwan China) to obtain tensile force-displacement curve of samples. The applied newton force was not pre-set, and the newton force was achieved from the experiment and recorded in the software. The newton force of the compression test of lattice structures were obtained at the same way as the tensile test. Besides, we added the machine information of the mechanical testing machine for compression test.

Comment 2: The XRD result of the composite materials needs to discuss more.

Author’s reply 2: Thanks for your suggestion. We have added more XRD discussion in the revision.

“After filled with CaCO3 or TCP particles, the broad hump from the XRD pattern of both PLA/CaCO3 and PLA/TCP became sharp and distinguished. Specially, peaks at 16.8° and 19.2°corresponded to crystal planes of (110/200) and (203) assigned to PLA polymer were detected in XRD patterns of composite filaments, which indicated a nucleation effect after incorporation of inorganic fillers. Besides, characteristic peaks of inorganic fillers were also observed. For example, strong peaks of CaCO3 at 29.5°, 35.9° and 39.4°were detected in composite filaments [23], confirming the successful incorporation of CaCO3.”

Comment 3: The ratio of PLA or TCP or CaCO3 is not clear in manuscript. Please mention clearly with weight mentioned as well.

Author’s reply 3: Thanks for your suggestion. To make the proportions more clearly, we have supplied the material compositions and the cold method in Table. 1 in the revised manuscript.

Comment 4: Authors emphasize fabrication of porous bioactive composite. Herein I suggest studying biocompatibility and cellular behavior on their product.

Author’s reply 4: Thanks for your constructive suggestions. We have studied in vitro cytocompability of the 3D-printed scaffolds. Live cell staining showed that cells adhered and proliferated well on the PLA/CaCO310% scaffolds. More detailed in vitro cytotoxicity and cellular behavior are in progress, and in vivo biocompatibility of 3D-printed PLA/CaCO3 and PLA/TCP scaffolds are also intended to be carried out in the future. We hope that we can provide more detailed data and information in our following manuscript.

    Once again, thank you very much for your thoughtful and thorough review. Hopefully, we have addressed all of your concerns. We look forward to receiving any information about our revision.

Round 2

Reviewer 1 Report

The authors should mention that FDM/FFF is a type of material extrusion additive manufacturing technique (according to ISO/ASTM terminology).

There should be more discussion on lattices that are fabricated using FDM/FFF instead of other processes such as SLM or SLA. 

Author Response

Author’s reply: Thanks for your suggestion. In the current revision, we added relevant introduction of lattice structure via FDM technique, and then explained the terminology of FDM and FFF according to ISO/ASTM: 2015 as material extrusion. Two additional reference were also added to support our modification.

Ref. [10] and Ref. [13]:

[10] Antony, S.; Cherouat, A., Montay, G. Fabrication and characterization of hemp fibre based 3D printed honeycomb sandwich structure by FDM process. Appl. Compos. Mater. 2020, 27, 935-953.

[13] David, P.; Ward, C.; Herrmann, G.; Etches, J. The manufacture of honeycomb cores using Fused Deposition Modeling, Adv. Manuf.: Polym.Comp. 2017, 3, 21-31.

    Once again, thank the reviewer very much for the professional comments and constructive suggestions, which are useful and helpful to improve our manuscript. Hopefully, we hope the modification meet your requirements.